# Deep learning predicts all-cause mortality from longitudinal total-body DXA imaging

Yannik Glaser [1 ✉], John Shepherd[2], Lambert Leong [2], Thomas Wolfgruber [2], Li-Yung Lui[3], Peter Sadowski [1] & Steven R. Cummings [3]

## Abstract

**Background** Mortality research has identified biomarkers predictive of all-cause mortality risk. Most of these markers, such as body mass index, are predictive cross-sectionally, while for others the longitudinal change has been shown to be predictive, for instance greater-than-average muscle and weight loss in older adults. And while sometimes markers are derived from imaging modalities such as DXA, full scans are rarely used. This study builds on that knowledge and tests two hypotheses to improve all-cause mortality prediction. The first hypothesis is that features derived from raw total-body DXA imaging using deep learning are predictive of all-cause mortality with and without clinical risk factors, meanwhile, the second hypothesis states that sequential total-body DXA scans and recurrent neural network models outperform comparable models using only one observation with and without clinical risk factors.

**Methods** Multiple deep neural network architectures were designed to test theses hypotheses. The models were trained and evaluated on data from the 16-year-long Health, Aging, and Body Composition Study including over 15,000 scans from over 3000 older, multi-race male and female adults. This study further used explainable AI techniques to interpret the predictions and evaluate the contribution of different inputs.

**Results** The results demonstrate that longitudinal total-body DXA scans are predictive of all-cause mortality and improve performance of traditional mortality prediction models. On a held-out test set, the strongest model achieves an area under the receiver operator characteristic curve of 0.79.

**Conclusion** This study demonstrates the efficacy of deep learning for the analysis of DXA medical imaging in a cross-sectional and longitudinal setting. By analyzing the trained deep learning models, this work also sheds light on what constitutes healthy aging in a diverse cohort.

## Plain language summary

Body composition – the overall proportion of fat, muscle, and bone in one's body – has been associated with mortality. It is important to better understand the relationship between body composition and mortality as changing body composition is an important goal of many drug and lifestyle interventions. Here, we combine medical images used for body composition measurement directly with information from the medical history of a large number of people to predict mortality. We use machine learning, which relies on mathematical models that extract useful features from images and use these to predict an outcome. Our findings show that combining body composition imaging with traditional mortality risk factors improves the prediction of mortality. This may help clinicians to more accurately predict who is at risk of dying in the future and target these patients with appropriate interventions.

[1] Information and Computer Sciences, University of Hawai'i at Mānoa, Honolulu, HI, USA. [2] University of Hawai'i at Mānoa Cancer Center, Honolulu, HI, USA. [3] San Francisco Coordinating Center, California Pacific Medical Center Research Institute, San Francisco, CA, USA. ✉email: yglaser@hawaii.edu

The field of mortality research has produced rich literature on how variables describing body composition can predict mortality. The Health, Aging, and Body Composition (Health ABC) study, is a prospective cohort study of 3075 individuals[1] that has provided many insights into the relationship between body composition and mortality. For instance, Newman[2] used the Health ABC dataset to show that baseline strength is predictive of all-cause mortality, while thigh muscle area obtained from computerized tomography (CT) scans and regional DXA lean mass showed no association. Building on these results. Santanasto and colleagues[3] used the dataset to show that change in muscle mass, derived from longitudinal CT scans, and greater than average weight loss are individually predictive of all-cause mortality. However, they failed to find any statistically significant relationship between mortality and visceral and subcutaneous abdominal fat or subcutaneous thigh fat. They also did not find a significant relationship between changes in lean and fat mass, derived from Total Body Dual-Energy X-ray absorptiometry (TBDXA) and all-cause mortality. Most recently, Westbury et al.[4] have analyzed the impact of baseline levels and change over time of various body composition variables on adverse health outcomes and overall mortality in the Health ABC dataset. Their results indicate that lower-than-average baseline levels are more predictive of adverse outcomes than greater decline over time.

In parallel, there have been efforts in related all-cause mortality prediction problems to include medical imaging data. Elton et al.[5] have successfully explored using abdominal CT scans for cardiovascular disease and five-year survival prediction, similarly, in another work focused on CT imaging, Yan and colleagues[6] used low-dose CT imaging to predict all-cause mortality for lung cancer subjects. However, such imaging-focused approaches have not yet been explored for general body-composition-based all-cause mortality prediction models.

So, while it is well-established that body composition and changes in muscle mass are predictive of mortality, changes in total lean and fat mass appear less relevant. Thus, mortality prediction models typically use only the baseline patient record. Furthermore, imaging data is not used directly by current body-composition-based mortality models—instead variables of interest are extracted for modeling, discarding all other information contained in the imaging.

Based on this knowledge, this study aims to test two hypotheses:

The first hypothesis is that risk features derived from raw total-body DXA imaging data using deep learning are predictive of all-cause mortality with and without clinical risk factors. TBDXA imaging scans contain rich body composition information such as central adiposity, regional muscle, fat, and bone mass, and bone density, making it the criterion method for body composition assessment[7]. Additionally, DXA scans are relatively cheap and widely available throughout North America and the world, can be taken at any age, and only use a low radiation dose making them well-suited to collect longitudinal data with[8]. CT, being another imaging modality used for body-composition-related mortality analyses and broadly used for diagnostic imaging, is comparably more expensive and higher in radiation dose[8]; coupled with the often limited availability of CT for risk screening due to its role as an emergency imaging modality, DXA-based modeling for mortality risk may be more immediately useful to clinical practice than modeling based on CT imaging. It is also easier to collect total-body scans with DXA than with other modalities that would otherwise be good candidates for this type of analysis, like MRI[9]. Two sample TBDXA scans with high and low energy channels separated can be seen in Fig. 1a.

The second hypothesis is that sequential TBDXA scans and recurrent neural network models (RNNs) outperform comparable models using only one observation with and without clinical risk factors. This hypothesis is supported by the previously mentioned work showing that some longitudinal changes in body composition are associated with mortality, together with the intuition that these and other changes in body composition should be reflected in the progression of longitudinal TBDXA scans.

Deep artificial neural network models (DNNs) are well-suited for this analysis, as convolutional architectures can capture translation invariances in imaging data, recurrent architectures can model variable length sequences of visit records, and multimodal architectures can combine multiple types of features such as images and tabular data. For example, Yala et al.[10] and Motwani et al.[11] have demonstrated the efficacy of deep neural networks for mortality prediction. Here, convolutional long short-term memory (LSTM) networks are employed to incorporate all available DXA images and clinical risk factors over multiple visits and capture longitudinal changes in body composition. This modeling approach is most closely related to the method proposed by Cui et al.[12] for Alzheimer's disease diagnosis from structural MRI. LSTMs are a type of RNN and have two major benefits: the model's ability to process sequences of different lengths without requiring padding or truncating any of the data, which is desirable because not all participants have the same number of datapoints available for various reasons; and RNN architectures model temporal dependencies of a sequence, unlike alternatives such as ensemble models. This also distinguishes the proposed method from previous efforts in the literature using longitudinal information such as Santanasto et al.[3] or Westbury et al.[4] who use feature-engineering (i.e. subtracting the baseline value of a marker from the current value to signify change over time in[4]) because the proposed approach models the time dependency directly using deep learning.

This study demonstrates that combining full TBDXA scans and traditional mortality risk factors results in stronger mortality prediction models than using either modality on its own. The results presented here also show that the progression of body composition and health markers over time can be leveraged to further improve mortality prediction, resulting in our best overall model achieving a 0.79 area under the receiver operator curve (AUROC) by integrating longitudinal TBDXA information and traditional mortality risk factors.

## Methods

**Study population.** The data used for our analysis was collected as part of the Health ABC study and we retrospectively analyze the completed cohort. Health ABC is a prospective cohort study of 3075 participants (48.4% men, 51.6% women) aged 70 to 79 years at the time of recruitment, 41.6% of whom are Black with the remaining 58.4% being non-Hispanic White. Participants were recruited from Medicare-eligible adults in metropolitan areas surrounding Pittsburgh, Pennsylvania and Memphis, Tennessee. Eligibility criteria were for participants to be within the age range; self-report having no difficulty walking a quarter mile, climbing 10 steps, or performing activities of daily living; have no history of treatment for cancer within the previous three years; and have no plans of moving out of the area within three years of being recruited[1,13]. All subjects signed an informed consent form and the IRB boards at each field center (University of Pittsburg, PA and University of Tennessee, Memphis, TN) approved the consent forms and protocol.

At intake, participants' medical characteristics were recorded through a questionnaire. Concerning this study, across the entire population, 58% self-reported a previous heart condition (one or multiple of: previous heart attack or myocardial infarction, a history of chest pain, previous congestive heart failure, previous strokes, a history of hypertension), 21% self-reported previous respiratory illness (one or multiple of: diagnosed with asthma,

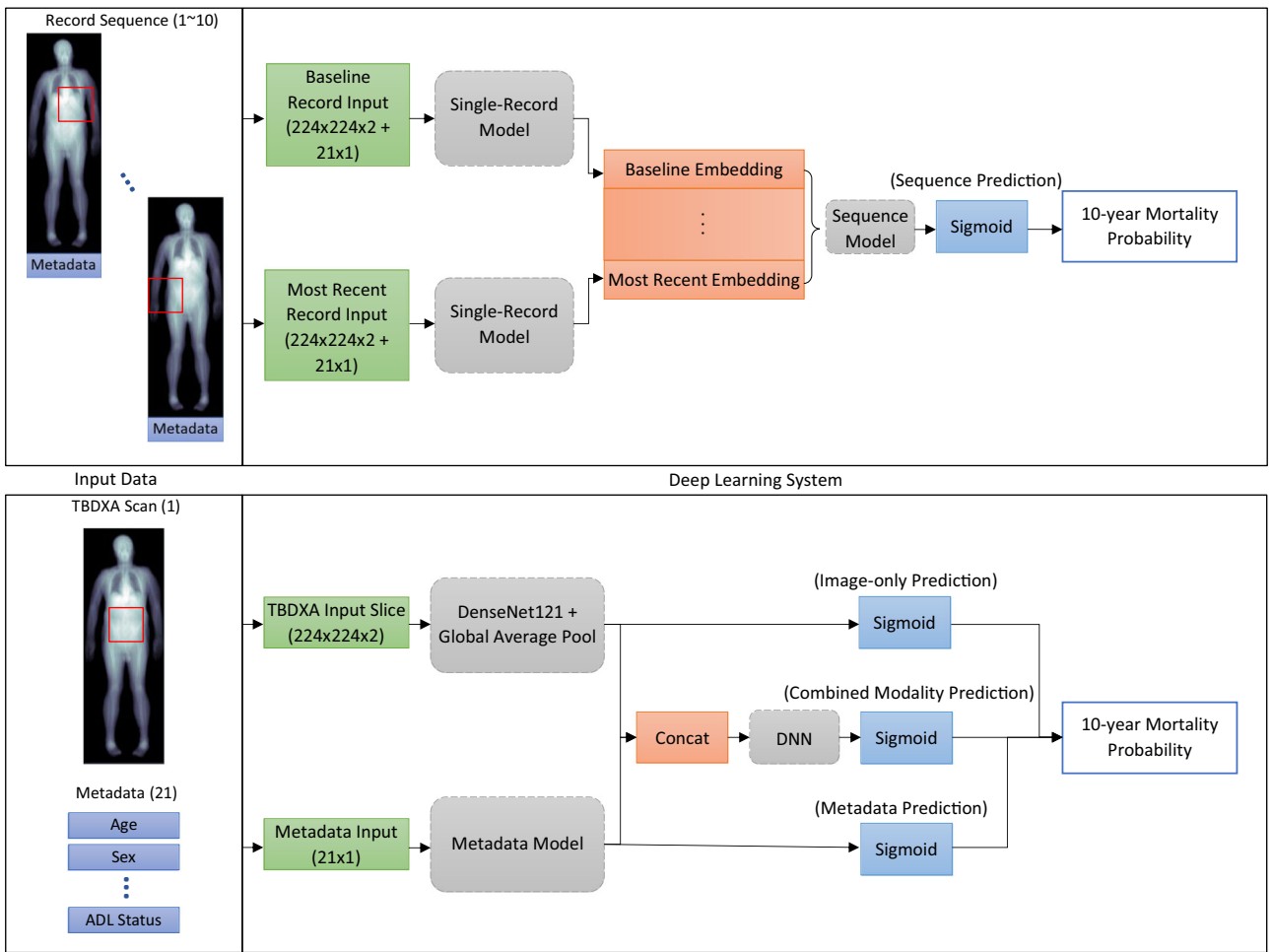

**Fig. 1 Schematic description of deep neural network models.** Single-record models (top). Predictions can be made either using each individual modality, or by combining embedding vectors from both modality models. Sequence models (bottom). Which modalities are used is determined by the single-record model which can be replaced to fit different input modalities. This model creates an embedding for all records in the sequence and then the resulting matrix is passed through the appropriate sequence model.

diagnosed with chronic bronchitis, diagnosed with emphysema, diagnosed with chronic obstructive respiratory disease (COPD), diagnosed with pneumonia in the past twelve months), 31% self-reported stomach or gallbladder issues (one or multiple of: previous stomach or duodenal ulcers, previous stomach or intestinal bleeding, previous surgery to remove parts of the stomach, gallstones), 15% self-reported being previously diagnosed with diabetes by a doctor, and 19% reported being previously diagnosed with cancer by a doctor.

After the conclusion of participant follow-up, 1992 deaths were recorded within the cohort.

**Clinical data**. Participants were followed for 16 years with regular check-ins consisting of a combination of questionnaire and exam measures. Clinical data collected from participants in the Health ABC study, hereafter referred to as "metadata", consists of demographics and anthropometric measurements (race, sex, age, height, weight, BMI), blood markers (blood glucose, fasting glucose, blood insulin, fasting insulin, hemoglobin A1c, interleukin 6), general indicators of fitness (walking speed over 3/4/6 m, 20 m, and 400 m; grip strength), and self-reported questionnaire answers (disability status for walking, climbing stairs, and activities of daily living; whether the participant had any recent falls and if so how many). Basic characteristics of this metadata can be found in Table 1.

**Image acquisition**. The Health ABC study attempted to collect whole-body DXA scans on eight different occasions[1]. These scans were collected using a Hologic system (Hologic QDR 4500, software version 8.21, Hologic Inc., MA) with strict acquisition procedures in place to ensure reproducibility, including a detailed DXA operations manual[14], annual operator training, and contracting the services of a DXA reading center[15]. Throughout the study, whole-body phantoms and human volunteers were used to verify proper DXA calibration[15].

**Data preparation and split**. Since questionnaire and exam measures were irregularly scheduled and did not always line up with each other or when imaging was done, not all data was collected contemporaneously with the scans. Which datapoints were included in the dataset used for this study was determined by when scans were collected. Missing values in other modalities were then simply backfilled with the most recent values available or a default out-of-distribution value.

The dataset was then split by participant with 70% of the being assigned to the training data set, 10% assigned to the validation split for hyperparameter tuning and early stopping during training, and 20% reserved as a hold-out test set for the final model comparison. This particular split was selected to ensure the maximal amount of training data for the DNN models while maintaining a representative test set, the validation data was only

**Table 1 Characteristics of study participant from a cross-sectional and longitudinal perspective across the entire dataset.**

|  | Cross-sectional mean (SD) | Mean (SD) change from first to last datapoint |
|---|---|---|
| Age, y | 75.2 (3) | 6.3 (3.1) |
| Height, m | 1.66 (0.1) | −0.006 (0.001) |
| Weight, kg | 74.8 (15.0) | −2.4 (6.2) |
| BMI, kg/m$^2$ | 27.1 (4.8) | −0.7 (2.3) |
| Fasting blood glucose, mg/dL | 102.9 (28.5) | 0.9 (24.3) |
| Blood glucose, mg/dL | 103.2 (29.1) | 0.9 (26.5) |
| Hemoglobin A1c, % | 6.0 (1.0) | −0.3 (0.8) |
| Interleukin 6, pg/mL | 3.3 (3.2) | 0.5 (3.0) |
| Walking speed over 4 m, 5 m, or 6 m, m/sec | 1.1 (0.3) | −0.2 (0.3) |
| Walking speed over 20 m, m/sec | 1.1 (0.2) | −0.2 (0.2) |
| Walking speed over 400 m, m/sec | 1.3 (1.1) | −0.1 (2.0) |
| Grip strength, kg | 27.6 (10.1) | −2.4 (6.6) |

a secondary priority where 10% of the dataset are a compromise to still get a reasonable performance estimate for hyperparameter selection and early stopping.

Categorical metadata features including race, sex, disability statuses, as well as whether the participant had any recent falls were encoded as one-hot vectors. All other numeric values were min-max-scaled between zero and one based on the training data distribution.

Using custom software developed by the authors in Python (version 3.8)[16], raw low- and high-energy (HE) X-ray attenuation values were extracted from the DXA scan file. For Hologic systems of this generation, low energy X-ray exposure conditions are 100 kVp (X mAs) with 6.1 mm equivalent aluminum (Al) filtration, and high energy X-ray exposures conditions are 140 kVp (X mAs) with 53 mm equivalent Al filtration. These raw attenuation images had a resolution (width and height) of $109 \times 150$ pixels, at 16-bit pixel depth. Each pixel had spatial dimensions of 2 mm×12.76 mm. All images were upscaled by a factor of two to a resolution of $218 \times 300$ using bicubic interpolation. Bone and soft tissues calibration phases in the scan file were used to restore the high and low images to their full resolution resulting in a width of 654 pixels. Image pixels were squared (2 mm × 2 mm) upon export by upscaling, via bicubic interpolation, by a factor of 6.38 in the y-direction for a final $654 \times 1914$ image.

**Single-record models.** Figure 1 shows the basic architecture of the single-record mortality models; input modalities are processed separately by appropriate architectures and the outputs can then either be concatenated for a combined-modality model or used separately for a final prediction.

Since the scans that are inputs to the image model have two "channels" (low and high energy attenuated scans), a 224 by 224 crop from the image is first passed through a single convolutional layer with a 1-by-1 kernel and three filters to conform with standard RGB (three channel) image dimensionality. The output from that convolution is then passed into a DenseNet121[17] model where the penultimate ReLU unit activations are mean-pooled to create a 1024-dimensional embedding passed to a single sigmoid unit for prediction in the image-only model. The DenseNet was initialized with ImageNet weights and training was done with a combination of the RAdam[18] optimizer and the lookahead[19] algorithm (also known as the "Ranger" optimizer). The first 20 epochs during training are a linear warmup from 0 to the maximum 0.001 learning rate and then linearly decreasing back for 50 epochs toward 0.00001. Lookahead has a 7-epoch synchronization period and a 0.4 slow weight update ratio. Training is stopped when no improvement in validation loss is observed for 10 epochs.

The metadata model is a neural network with a single 32-unit, ReLU activation hidden layer and a single sigmoid unit for the final prediction. This model is trained using the Adam optimizer[20] with the suggested parametrization until no improvement in validation loss is observed for 10 epochs.

The combined-modality model concatenates the 1024 and 32 dimensional embedding vectors from the two single-modality models into a vector that is then passed through a 512-unit and a 64-unit fully-connected ReLU layer and finally a single sigmoid unit for mortality prediction. This model is trained in two steps. First, the weights for the two modality subnetworks are locked and only the last three dense layers are trained using the Adam optimizer for 30 epochs, training is stopped early when no validation loss improvement is seen for 10 epochs. Second, all weights in the network are unlocked and it is trained using the Ranger optimizer in the same configuration as above but with a 12 epoch warmup portion. Training is concluded after not improvement in validation loss is seen for 10 epochs.

During training, a 50% dropout rate is applied to all outputs from the penultimate model layers before the sigmoid unit to avoid overfitting. Additionally, an image augmentation scheme is applied during training for all models with that input modality. First, to fit the height and width input dimensionality of the pretrained DenseNet, a random 224 by 224 section of the image is cropped out. Then four possible augmentations may be applied to the crop, each with an individual probability of 30%: a blur with a random kernel size between 3 and 5, the crop may be rotated up to 10 degrees clockwise or counterclockwise, a Gaussian noise kernel with mean 10 and a random variance between 1 and 20, up to three Cutout[21] dropouts of up to one-third of the image each.

All models are evaluated by their AUROC score on the test set. For all models involving scans, instead of the random cropping done during training, the 224 by 224 scan slice is obtained by always center-cropping and no augmentation is done. Records were assigned a positive (1) label if the person died within 10 years of when the scan was taken and otherwise a negative (0) label.

All confidence intervals shown across all results are calculated by computing the AUROCs of 1000 bootstrap samples from the test set.

**Sequence models.** For each modality, the single-record models serve as a base model. Each participant is represented as a sequence of visits where data from each visit is passed through the appropriate model to create an embedding, resulting in a sequence of embeddings after all visits are processed. For instance, for the combined-modality sequence model, the scan and metadata obtained at each visit are first passed through the combined-modality single-record model with the final sigmoid unit removed, resulting in a 64-dimensional embedding vector for

each visit. To each vector, one additional number is appended indicating the time passed between this visit and the previous one in days scaled to be between −1 and 0; so, the first vector always has a zero appended and the ones after some negative value because of the scan ordering. Thus, this sequence of vectors represents the entire sequence of visits for a participant, starting with the most recent scan and going back in time. This sequence is the passed through the sequence model, whose architecture is consistent for all modalities. The embedding sequence is passed through a recurrent layer consisting of 8 LSTM cells[22] followed by a subsequent four fully-connected 64 unit ReLU layers and a single sigmoid output unit. Again, during training a 50% dropout rate is applied before the output unit.

While training the LSTM parameters, the single-record models' weights are locked throughout and at no point change. Training is done until no improvement in validation loss is observed for 30 epochs. The standard Adam optimizer is used and for the image models the same augmentations are used as for the single-record models. Additionally, during training, there is a 10% probability for each visit recorded for a participant to be dropped from the sequence, potentially shrinking the sequences to a minimum of two visits at which point no further visits will be dropped. This is to further avoid overfitting. Specifically, this measure is taken to avoid the fitting to sequence length information. It is easy to imagine the model otherwise making the connection that if a participant has significantly less than the maximum number of scans, they are likely to have died before the conclusion of the study and should thus be assigned a higher mortality risk.

All models use the binary crossentropy loss for training and are implemented in Python 3.8[16] using the Tensorflow package[23]. Details on hyperparameter tuning for all models is provided in the Supplemental Information including tuning procedure and detailed hyperparameter ranges in Supplementary Table 1.

Similar to the single-record models, all models are evaluated by their AUROC score on the test set, again using only center-crops for any models using imaging data. Since each participant is now a singular datapoint consisting of a sequence of records labels were assigned based on the most recent record in the sequence, where a death within 10 years of that record is assigned a positive (1) label and everything else is assigned a negative (0) label.

**Subgroup performance**. Subgroup performance is established on the test set alone. The race and sex subgroups are determined based on the demographic information collected as part of the study. The cause-of-death subgroups are determined based only on the combination of participants who died within the follow-up window of the Health ABC study and for whom cause-of-death information was available, serving as positive examples, and participants who lived either until the study follow-up period concluded or for at least 10 years after their last available scan, serving as negative examples. For the purpose of Table 2 in the main text, to determine both cardiovascular and cancer deaths, either cancer or cardiovascular disease had to have been determined to be either the immediate or underlying cause of death. The diabetes subgroups are determined based on the participants' fasting glucose levels where less than 100 mg/dL were considered "normal", between 100 and 150 mg/dL were considered "prediabetic", and anything above 150 mg/dL was considered to belong to the "diabetic" subgroup. For the BMI subgroups, the common BMI categories are applied to divide the participants (<18.5: "underweight", 18.5–25: "normal", 25–30: "overweight", >30: "obese").

**Feature importance**. For the ablation study, all architecture and training hyperparameters are identical to what is detailed for the full models with all inputs.

**Table 2 Model performance breakdown.**

| | Full test set | | | Participant race | | Participant sex | | Cause of Death | |
| --- | --- | --- | --- | --- | --- | --- | --- | --- | --- |
| | Overall (n = 3720) | Baseline scan (n = 639) | Most recent scan (n = 639) | Black (n = 1378) | White (n = 2342) | Female (n = 1880) | Male (n = 1840) | Cardio-vascular disease (n = 682) | Cancer (n = 577) |
| Image Model | 0.63 (0.62–0.65) | 0.63 (0.58–0.68) | 0.64 (0.59–0.68) | 0.63 (0.60–0.66) | 0.65 (0.62–0.67) | 0.60 (0.58–0.63) | 0.61 (0.59–0.64) | 0.67 (0.65–0.70) | 0.66 (0.63–0.69) |
| Metadata model | 0.69 (0.68–0.71) | 0.67 (0.63–0.71) | 0.73 (0.69–0.77) | 0.66 (0.63–0.69) | 0.71 (0.69–0.73) | 0.67 (0.64–0.69) | 0.69 (0.67–0.71) | 0.74 (0.72–0.77) | 0.69 (0.66–0.71) |
| Combined-modality model | 0.71 (0.70–0.73) | 0.68 (0.64–0.72) | 0.74 (0.70–0.78) | 0.69 (0.66–0.71) | 0.73 (0.70–0.75) | 0.68 (0.65–0.71) | 0.72 (0.70–0.74) | 0.76 (0.73–0.78) | 0.70 (0.67–0.73) |
| Image sequence | 0.73 (0.63–0.75) | - | - | 0.72 (0.65–0.78) | 0.72 (0.67–0.77) | 0.67 (0.62–0.73) | 0.72 (0.66–0.78) | 0.77 (0.71–0.81) | 0.72 (0.66–0.78) |
| Metadata sequence | 0.76 (0.71–0.78) | - | - | 0.71 (0.64–0.77) | 0.77 (0.72–0.82) | 0.71 (0.65–0.76) | 0.75 (0.69–0.81) | 0.81 (0.76–0.86) | 0.73 (0.66–0.79) |
| Combined-modality sequence | 0.79 (0.75–0.82) | - | - | 0.76 (0.70–0.82) | 0.80 (0.75–0.85) | 0.75 (0.69–0.80) | 0.80 (0.75–0.85) | 0.83 (0.78–0.87) | 0.77 (0.72–0.83) |

Area under the receiver operating characteristic (AUROC) and 95% confidence intervals for all models on full test set and subpopulations.

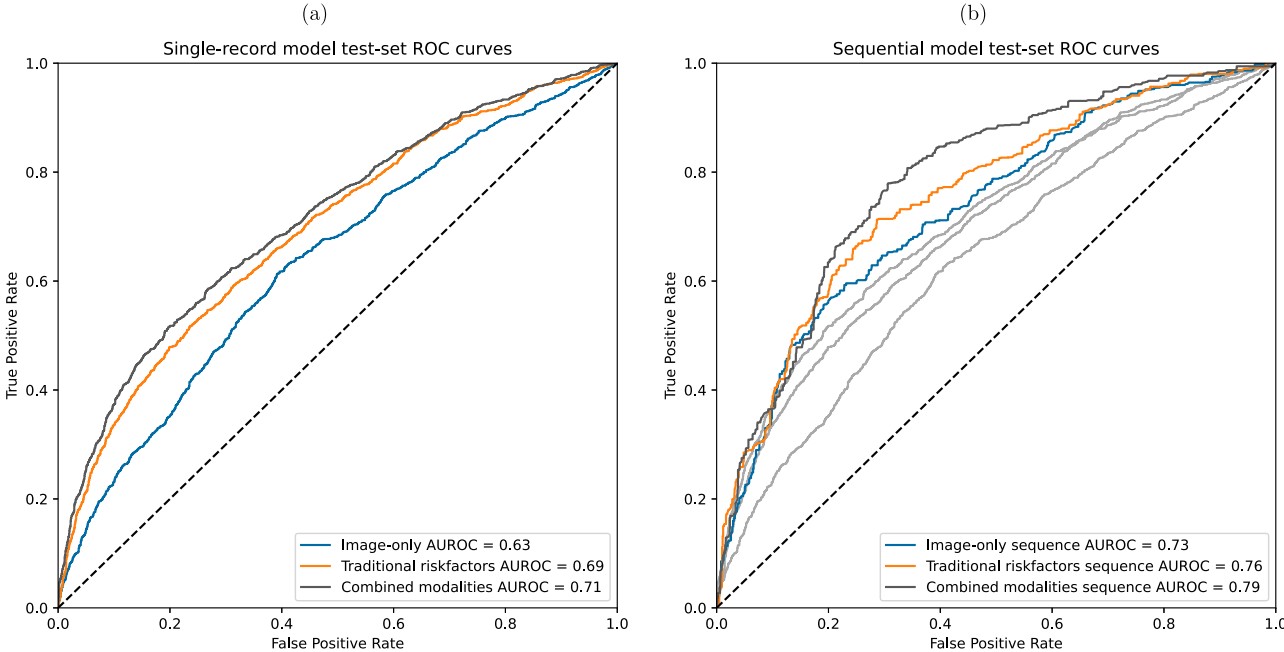

**Fig. 2 Model performance comparison. a** Single-record model receiver operating characteristic (ROC) curves and area under the receiver operating characteristic (AUROC) on held-out test set. **b** Sequence model ROC curves and AUROCs on held-out test set.

Saliency maps are obtained using both the integrated gradients[24] and Grad-CAM[25] methods. Since the image model only considers a 224 by 224 sections at any time a 224 by 224 sliding window is used to get overlapping crops from the scan. For each crop the sliding window is advanced 50 pixels first in x direction and when the edge of the scan is reached in y direction until each part of the scan has been covered in at least one slice. After activations for all crops are obtained, they are overlayed and averaged to get the final map over the entire scan. To obtain the gradients for the integrated gradients maps, the integral is approximated over 128 steps and an all-zero baseline vector.

## Results
### Single-record models
*General performance.* The first set of models trained are the "single-record models", where the dataset was treated as cross-sectional and each scan represents an independent sample. Figure 2a shows the test set AUROC of three DNNs with different model inputs trained in this setting.

The baseline model uses only metadata: a mix of blood markers, questionnaire answers, and other general health markers collected from patients. This model is a fully-connected DNN with a single hidden ReLU layer and achieves a test set AUROC of 0.69.

In contrast, the model combining the metadata input with the TBDXA imaging data outperforms the metadata model with a 0.71 AUROC on the test set. This model concatenates representations from the metadata model and an image-only DenseNet[17] model whose input is the two-channel TBDXA scan. The resulting learned feature vector is then passed through additional densely-connected layers for the final prediction (Fig. 1). The image-only model on its own has a 0.63 AUROC score.

*Alternative evaluation criteria.* To allow for a fair comparison with the sequence models down the line, the single-record models were also evaluated on only the most recent scan for each participant, resulting in predictably higher AUROCs for all three models (image −0.64, metadata −0.73, combined-modality −0.74) (Table 2) relative to the unrestricted test set. The intent of this adjustment was to match the test sets for the sequence and single-record models as closely as possible. By using only the most recent scan for each participant for the single-record models, it is ensured that 10-year mortality labels are the same for both models across the test set, whereas otherwise the same participant can have their label change throughout the course of the study if they died within follow-up but more than ten years after their baseline scan. In an effort to match common practice in mortality analysis[2], the single-record models were also evaluated on only the baseline records for each participant and the results were also included in Table 2 as well as the Supplementary Table 2 for more subgroups. It is however worth emphasizing that this does not present a fair comparison of the different approaches because label distributions are different between the sequence and single-record testing data in this set-up.

### Sequence models
*General performance.* The next set of models are the "sequence models". For these models, the dataset was treated in a longitudinal manner, where a single datapoint consists of all scans collected for each individual study participant in chronological order from the baseline to the most recent scan available. Three different RNNs were trained, again one for each input modality and performance is shown in Fig. 2b.

As with the single-record models, the image-only model performance is the lowest (0.71 AUROC) with the widest 95% confidence interval, followed by the metadata (0.76 AUROC) and combined-modality (0.79 AUROC) models, both with tighter confidence intervals (Table 2). For the purpose of training and evaluating these models, the predicted 10-year mortality is based on the most recent scan collected for each individual in the test set.

*Sensitivity and specificity in the strongest models.* In an effort to better understand the combined-modality sequence model performance, the sensitivity and specificity scores of the two strongest sequence models—the metadata-only and the combined-modality

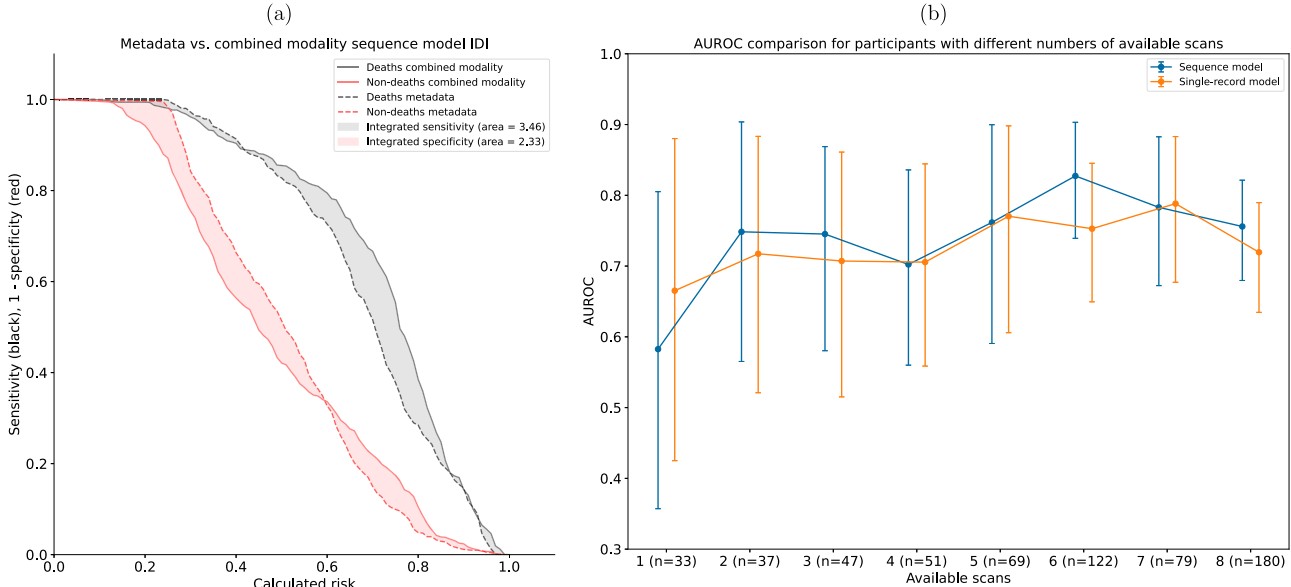

**Fig. 3 Sequence model performance details. a** Risk assessment plot for the performance comparison between the metadata sequence model and the combined-modality sequence model shows overall improvement in sensitivity and specificity for the model including TBDXA scans. The combined area between the death and non-death curves for the models represents the integrated discrimination improvement (IDI) score. A positive IDI means that, generally, risk scores increased for patients that died within the 10 year window, while for patients that did not die risk scores assigned by the combined-modality model decreased compared to the metadata model. In the plot, the death curves (grey) are similar to plotting sensitivity across all risk scores, while the non-death curves illustrate 1—specificity overall risk scores. Accordingly, to show that a model has high sensitivity, it is desirable for the grey curve to be high and toward the right of the plot, while a model with high specificity will have a red curve that is low and toward the left of the plot. For most risk scores, the combined-modality model is on par with or better than the metadata model in terms of both sensitivity and specificity; only for risk scores above ~60% does the metadata model have improved specificity. **b** Plot comparing the performance of the sequence model and single-record model for participants with different numbers of available scans. The single-record model is evaluated only on the most recent scan while the sequence model is evaluated on all available scans. Error bars show 95% confidence intervals calculated using bootstrapping on each respective subset.

models—were compared in detail. The analysis, illustrated in Fig. 3a, shows that for a large range of risk scores, the combined-modality sequence model has improved sensitivity and specificity, which translates to an integrated discrimination improvement (IDI)[26,27] of 5.79 (3.46 for events and 2.33 for non-events) and category free net reclassification improvement (cfNRI) of 51% on the test set. The cfNRI is the unweighted sum of the net proportion of events assigned a higher risk score (cfNRI for events being 25%) and non-events assigned a lower risk score (cfNRI for non-events being 26%), meaning that for both events and non-events the combined-modality sequence model assigns more accurate risk scores about 25% more often than it assigns more inaccurate risk scores relative to the metadata-only sequence model.

*Impact of sequence length on model performance.* To understand how sequence length affects the sequence model's performance, participants in the test set were split into subgroups based on how many scans were collected for each as part of the study (i.e. all participants with only one available scan were grouped, all with two available scans, and so on), and then the combined-modality single-record model and sequence model were evaluated separately on each subgroup, with the single-record model's performance being based on its prediction for the most recent available scan. Figure 3b shows the results of this analysis, where the sequence model is outperformed only for the subgroup with just one available scan and is otherwise consistently similar or better in performance than the single-record model.

**Subgroup performance**. In addition to evaluating the models on the entire dataset, analyzing subgroup performance can also shed light on strengths and weaknesses of our approach; of particular interest might be the performance for different sexes, as many

traditional approaches[28] create separate models for these groups outright. Table 2 includes model performance for subgroups based on participant race, sex, and cause-of-death. Performance trends observed across the entire test set still largely persist here, with the combined-modality models outperforming their single-modality counterparts and the sequence models performing better overall. The subgroup for which all models exhibit the best performance is for participants who died of causes related to cardiovascular disease, with the combined-modality sequence model achieving the highest AUROC of 0.83. More detail on the cause-of-death subgroup analysis is provided in the Supplementary Table 3, where the model performances for the four most common primary and underlying causes of death are recorded respectively.

**Feature importance**
*Ablation study results.* To get a better understanding of what features are most useful to the models, an ablation study was conducted where one metadata feature was removed at a time and models were trained and evaluated without access to that feature. Other studies[10] have used gradient-based methods to attempt to gain insights into feature importance, but while these methods are easy to use, they do not necessarily find the most relevant features[29,30]. It is also worth noting that while omitting the image as an input to the model is not explicitly included in the ablation results, the resulting model would be identical to the metadata model whose performance is detailed in Table 2.

The ablation results for the combined-modality single-record model are depicted in Fig. 4a. The figure illustrates that the two most important metadata variables for model performance are walking speed over a medium-length distance (20 m) and sex, both with a 0.01 reduction in AUROC. The metadata variable

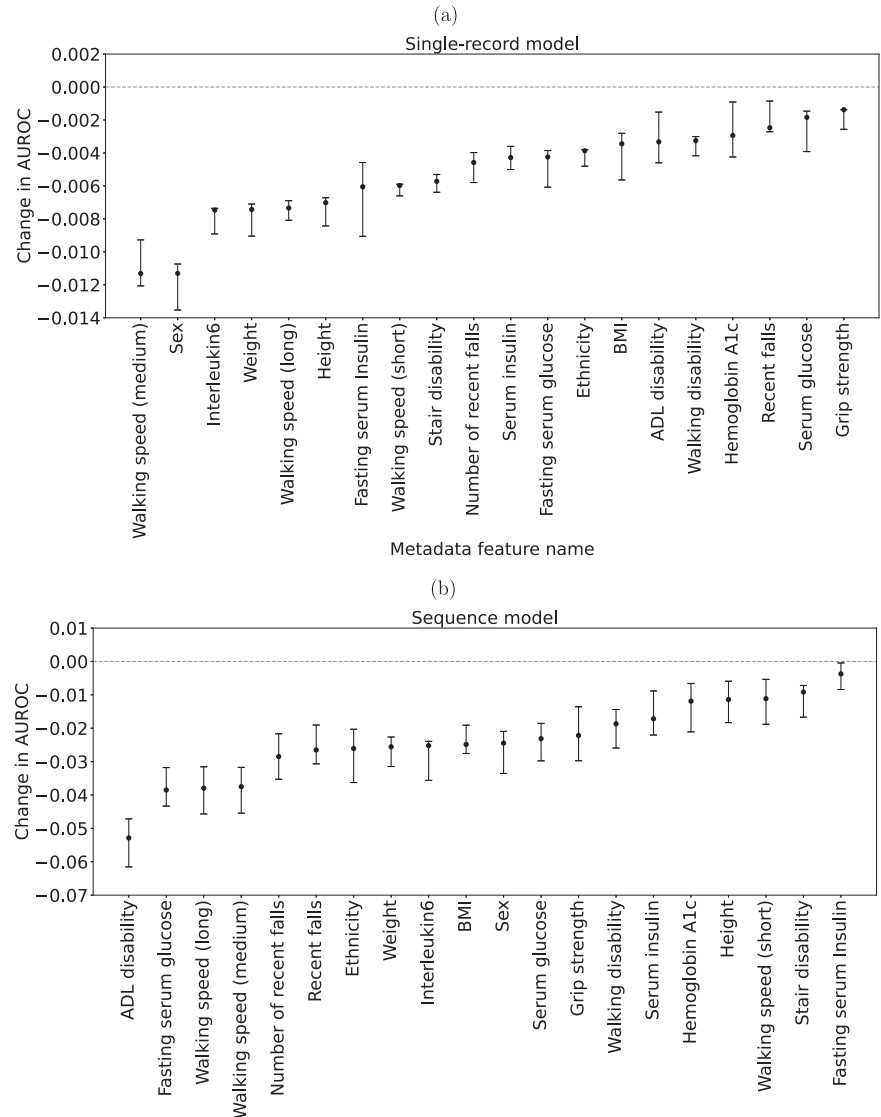

**Fig. 4 Feature importance scores.** Scores are change in area under the receiver operating characteristic (AUROC) compared to the full model and error bars show 95% confidence intervals calculated using bootstrapping. **a** Ablation study results on test set (n = 3720) for single-record model. **b** Ablation study results on test set (n = 639) for sequence model.

with the least impact on model single-record model performance is grip strength resulting in only a 0.002 drop-off in AUROC when ablated.

Likewise, Fig. 4b depicts the ablation study results for the combined-modality sequence model. Here we observe larger performance drops, specifically for ADL disability status (−0.05 AUROC), fasting serum glucose and walking speed over long (400 m) and medium (20 m) distances (all decreasing AUROC by roughly 0.04). The least impactful variable is fasting serum insulin with a 0.005 AUROC drop.

*DXA feature maps.* While the ablation study is designed to provide insight into the contribution of metadata variables, saliency maps were generated for the combined-modality single-record model in an effort to understand the specific contribution the TBDXA scans make to the model's performance. Two specific methods for feature visualization were selected because their different properties grant insights into different properties of the model. Figure 5a shows pixel-wise integrated gradients[24] visualizations. The maps show each pixel's contribution to the model's

ultimate output prediction for both a high-risk case and a low-risk case. Figure 5b shows Grad-CAM[25] visualizations for the same two cases. Grad-CAM feature maps visualize higher-level features in the network. Instead of allowing for pixel-level attributions, these maps illustrate for which regions of the image convolutional layers in the network show the greatest activation.

## Discussion

The goal of this study was to test two hypotheses ultimately aimed at improving all-cause mortality prediction models. The first hypothesis tested stated that raw TBDXA imaging data can be used by deep learning models to add predictive power to all-cause mortality models. Our single-record model comparison shows that a model with access to both metadata variables and TBDXA imaging outperforms a model using either modality individually, demonstrating that TBDXA indeed contains complementary information to traditional mortality risk fact and that deep learning can effectively extract this information to improve prediction of all-cause mortality. If, on the other hand, no additional information relevant to mortality prediction were present in the

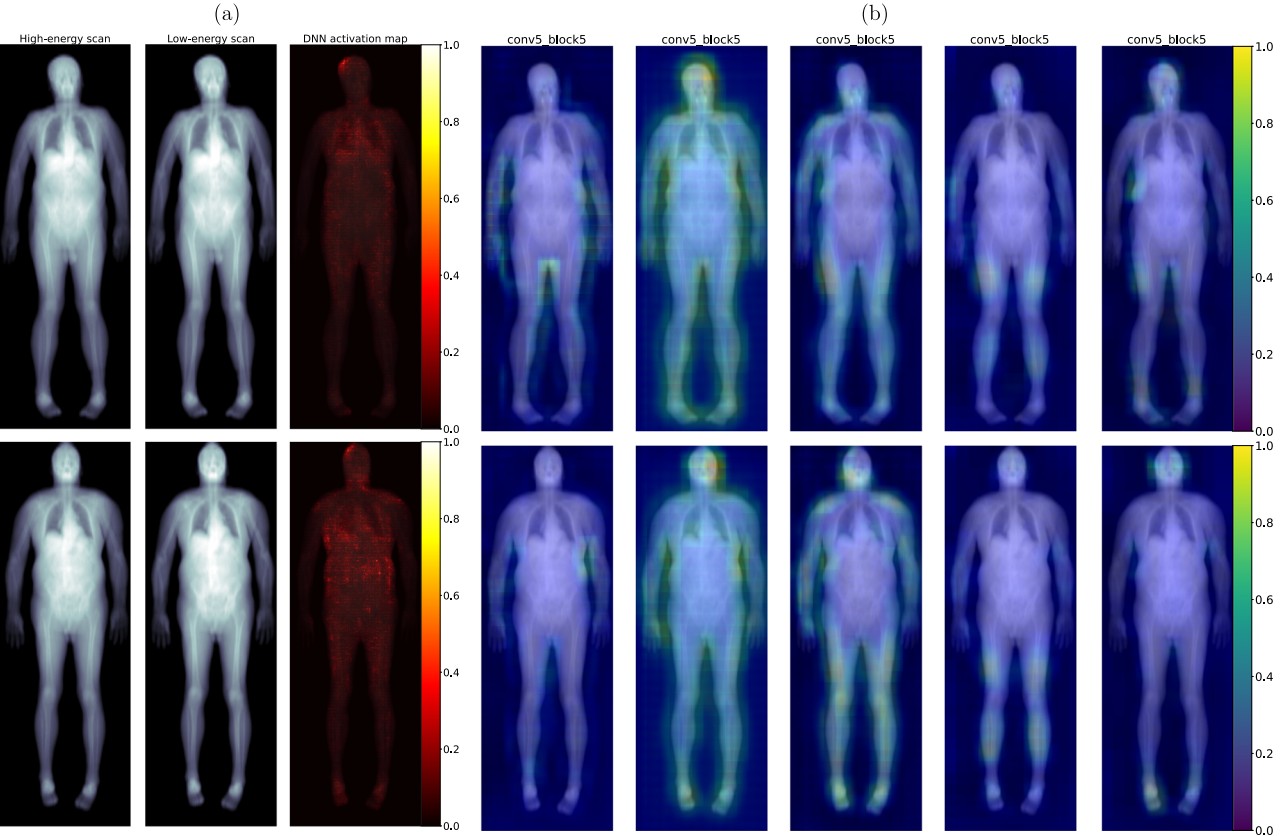

**Fig. 5 Saliency maps. a** Integrated gradients pixel-level saliency maps. Low mortality (top row) and high mortality (bottom row) sample scans from test set. Participants are 73 year old White males with similar body measurements. Rows from left to right depict: high-energy scan, low-energy scan, single-record combined-modality saliency maps overlayed on scan. **b** Grad-CAM higher-level convolutional feature maps for the same participants depicted in **a**. Feature maps depicted are all from convolutional layers within the dense block of the combined-modality single-record model before the last pooling layer.

scans, the model could at best match the performance of the metadata-only model and not exceed it. Other work in this area has already shown that some DXA-derived body-composition measurements can be predictive of all-cause mortality. Westbury et al.[4], for instance, showed appendicular lean mass and total-body fat mass, both derived from TBDXA, to be all-cause mortality predictors, while recent work by Laddu and colleagues[31] shows that DXA estimates of central adiposity adds predictive power to all-cause-mortality models in postmenopausal women. Farsijani et al.[32], in an effort to compare CT- and DXA-derived measurements for mortality prediction, find DXA fat mass to be predictive of all-cause mortality. The authors of this work also propose a novel approach based on compositional data analysis to derive more sophisticated body-composition based predictors from imaging, resulting in the finding that lower DXA-derived fat mass versus lean mass is also predictive of mortality. This approach can be seen as a limited effort in the same direction as our proposed method of deriving richer predictors from DXA imaging data. However, our method of training a deep learning model to extract information from the raw TBDXA scan employs more powerful models that can extract information contained in TBDXA imaging to automatically derive complementary features to what is already represented in traditional risk factors. This end-to-end approach removes the reliance on crude summary statistics such as fat or lean mass which must be derived from images by a radiologist or segmentation algorithm.

The second hypothesis tested stated that RNN models using TBDXA scans collected over multiple visits for an individual will outperform models that use the same information from only a single visit. Our sequence model performances show that

a model with access to longitudinal data performs better than a model with access to only a single patient record. This observation held for all modalities, where all sequence models outperformed their respective single-record counterparts, providing strong evidence that integrating longitudinal information in mortality prediction models is beneficial. To our knowledge, this observation is novel in the all-cause mortality literature as no such complicated multivariate model has been assessed on longitudinal data before our study. Previous studies conducted on longitudinal data for all-cause mortality prediction have demonstrated a complicated relationship, especially for DXA-derived predictors. Some predictors, for example grip strength[33] and walking speed[34], are well established to have a strong longitudinal association with all-cause mortality. However, for instance Santanasto et al.[3] showed that longitudinal changes in lean and fat mass alone are not related with all-cause mortality when adjusting for potential confounders. In light of the results presented in this paper, this suggests that simple summary values derived from TBDXA such as total lean and fat mass may not be sufficient to represent the complicated longitudinal information contained in a sequence of DXA scans. Interesting for our observed performances is the work by Westbury et al.[4] presenting results where baseline values were found to be generally more predictive than longitudinal changes in predictors. The authors of that work suggest incorporating both, baseline values and change over time, into all-cause-mortality models, which is implicitly accomplished by our RNN. Our results also show a greater relative improvement from TBDXA inclusion in the longitudinal setting versus the single-record setting, suggesting that unlike the risk factors analyzed in

Westbury et al.[4], the raw imaging data may have more predictive power in a longitudinal context.

Overall, this study demonstrates that risk factors derived from TBDXA using deep neural networks can supplement known mortality risk factors, and that changes in body composition over time can be a stronger predictor of mortality than any observations collected during a single visit.

**Subgroup performance**. The subgroup performance shown in Table 2 is critical to evaluate the stability of our models for different populations. Overall performance trends are very consistent, with the combined-modality models performing the best for both the single-record and sequence scenarios, and the sequence models outperforming the single-record models overall. Some other trends also stand out in this analysis.

When analyzing the most common causes of death recorded in the dataset, cancer and cardiovascular disease, the models performed most strongly on the group with cardiovascular disease. In fact, every single model achieved the highest AUROC observed for any subgroup of the testing data for the participants who died of cardiovascular disease. This observation aligns well with the existence of body-composition-related factors contributing to cardiovascular disease, information that would be available to the model through both the TBDXA and metadata variables.

Furthermore, the analysis shows that the models perform better on White participants than on Black individuals, which is possibly a result of the data distribution being skewed (58.4% White). The fact that the models also have better AUROCs for the male subset of the data than for the female is in line with previous analysis that has shown that certain body composition trends known to affect mortality appear to be more pronounced in men than in women[3,35]. It bears pointing out also, that while other methods commonly split the dataset on sex and develop separate models for men and women[1,4], our deep learning approach uses a single model for both groups. The model is flexible enough to learn differences between groups but can benefit from the larger training dataset that results from combining groups. More performance details on subgroups based on baseline diabetes status and baseline BMI are included in the Supplementary Table 4. Overall, the performance trends observed above still consistently hold for those subgroups as well.

**Ablation study**. The ablation study results grant insight into what metadata variables are especially predictive of mortality in the combined-modality setting where imaging data is also available to the model. For the single-record model, the results mostly show that no single variable has an outstandingly large impact on model performance, which is not surprising given that many of these variables are correlated with each other as well as the TBDXA scan. Nevertheless, the sequence model shows more pronounced performance drops when certain variables are not available. ADL disability status, fasting serum glucose, and walking speeds stand out by decreasing performance by more than 0.04 AUROC each when ablated. This relatively large drop in performance indicates that the longitudinal progression of these variables contains a large amount of useful information for mortality prediction. Performing survival analysis on the dataset provides some hints as to why these variables might hold more weight to the model. Participants whose disability status either changed from not being disabled to being disabled over the course of the study or who were classified as disabled from the start had shorter survival times than their peers (Supplementary Fig. 1). Similarly, for walking speed, people in the bottom quartile (i.e., slow walkers), whose walking speed decreased the most over the course of the study, had a much lower median survival time.

While the survival discrepancies are less severe for fasting glucose groups, there still is a clear trend where participants who started the study as prediabetic and either became diabetic or remained within that group had shorter survival times than the remaining participants with lower fasting blood glucose levels. More survival analysis related to the ablation results can be found in the Supplementary Fig. 1.

**DXA feature maps**. The saliency maps in Fig. 5 highlight regions of the DXA scans that are of particular value for the mortality models' predictions and can serve as a starting point when analyzing what parts of the TBDXA imaging drive the increase in mortality prediction performance. While it is not sensible to derive precise conclusions from these visualizations, they do grant some insight into deep convolutional models which are otherwise often considered a "black box".

At a low level, the pixel-wise integrated gradient maps especially highlight a trend in our models where no specific region or feature obviously stands out. This suggests that beyond what is reflected in tabular risk factors, diffuse (or disseminated) changes in tissues, rather than one single area such as the trunk area and central abdominal adiposity, contribute to the models' mortality prediction. This may also be a consequence of modeling choices. Because models, both during training and test time, only get access to 224 by 224 subsections of the image, it is impossible for the model to pick up on global features like overall body shape or posture, whereas the models are actively incentivized to learn "universal" features that are effective at predicting mortality regardless of what body part the slice is from.

At a higher semantic level, the Grad-CAM[25] maps help shed further light on image model behavior. By visualizing high level features in the model, one can observe convolutional layers that show activation across the entire scan (for instance the second column in the figure) and others that seem more specialized and only activate for specific regions in all scans. This allows for some more speculation as to what components of the scan are especially informative to the model, however, no conclusions should be reached based on these saliency maps alone without further investigation. Many of these maps—for instance columns one and three—show heightened activation around the periphery of the body, perhaps identifying body shape or subcutaneous fat distribution. Column four on the other hand shows the strongest activation in the femur area.

Overall, these maps seem to suggest that a lot of the information derived from scans to complement the metadata might be localized body composition information, distribution of lean and adipose tissue, or even bone shape and degeneration. A set of Grad-CAM maps for all convolutional layers in the last dense block of the model is provided in the Supplementary Fig. 2.

Most importantly, the fact that both the integrated gradients and Grad-CAM maps show activation in semantically meaningful regions serves as a sanity check to ensure that the models do not exploit information contained in the images that is not biologically meaningful.

**Limitations and future direction**. While the results presented here are encouraging and make us confident that our method can further advance mortality research, there are several limitations that need to be addressed in future work. While the Health ABC dataset is relatively large and diverse, it has limitations, especially for deep learning models. Our analysis needs to be extended to more diverse ethnicities and populations with larger age ranges.

Because the dataset is relatively small for training a large convolutional neural network, we had to deliberately use aggressive cropping as a form of augmentation during training,

limiting the model's "field of view" for any given example, and preventing it from fitting to global information like pose and overall tissue distribution. Given access to a larger dataset, it is reasonable to assume that training a similar model but with the full TBDXA scan as an input could yield further performance improvements and valuable insights into what drives mortality.

An additional complication is that the last scheduled Health ABC exam to include TBDXA scans was in 2006 while follow-up concluded in 2013. This means the mortality labels are noisy for later scans, because there might have been unrecorded deaths within the 10-year window after the last scan. This introduces an unintended bias in the sequence model where, as sequence length increases, the model becomes more likely to assign a lower mortality risk. Our analysis in Fig. 3b shows that the sequence model is more accurate than the single-record model even for the early years of the study where this effect is not a factor, but this subtlety must be considered when evaluating any model of longitudinal data.

We plan on addressing these concerns in future work by expanding our analysis to other datasets, gaining results on more diverse data, and increasing confidence in our models. Additionally, we want to repeat this type of analysis for CT imaging, which has also been collected as part of the Health ABC study. CT-based models could be used for opportunistic screening[36] in the clinical workflow and comparing these models with the ones presented in this work could shed further light on drivers of mortality and improve risk modeling overall.

## Data availability

The data that support the findings of this study are available from the National Institute on Aging, but restrictions apply to the availability of these data, which were used under license for the current study, and so are not freely available. Data however can be requested through the study's website at healthabc.nia.nih.gov.

## Code availability

All code for analysis as well as running the trained models is available at github.com/hawaii-ai/tbdxa_mortality and Zenodo[37].

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

## Acknowledgements

The authors would like to acknowledge the participants of the HealthABC study. This research was supported in part by the Intramural Research Program of the National Institutes of Health, National Institute of Aging. Technical support and advanced computing resources from the University of Hawaii Information Technology Services—Cyberinfrastructure are gratefully acknowledged.

## Author contributions

Developed the concepts and designed the study: S.R.C., J.A.S.; T.B.D.X.A. processing software development: J.A.S., T.W., L.L.; data storage, upkeep, and curation: L.Y.L., L.L., T.W.; machine learning modeling and statistical analysis: Y.G., P.S.; manuscript drafting and revision: all authors; approval of final manuscript version: all authors; agrees to ensure questions regarding accuracy or integrity of the work will be appropriately resolved: all authors.

## Competing interests

The authors declare no competing interests.

## Additional information

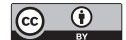 ns license, unless indicated otherwise in a credit line to the material. If material is not included in the article's Creative Commons license and your intended use is not permitted by statutory regulation or exceeds the permitted use, you will need to obtain permission directly from the copyright holder. To view a copy of this license, visit http://creativecommons.org/licenses/by/4.0/.

© The Author(s) 2022

