## [Peer Review File · Communications Medicine]

Reviewers' comments:

Reviewer #1 (Remarks to the Author):

The present study "DEEP LEARNING IDENTIFIES BODY COMPOSITION CHANGES OVER TIME IN TOTAL-BODY DXA IMAGING TO PREDICT

ALL-CAUSE MORTALITY" aims to demonstrate that the assessment of body composition by DXA can improve mortality risk prediction of a deep neural network. For this the authors use two approaches. The first approach, the single record model, aims to predict 10-year mortality probability from a baseline DXA scan that is processed with a convolutional neural network (DenseNet121) and after global average pooling gets concatenated with the metadata-model, a small network that processes known risk factors of 10-year mortality. The second approach takes advantage of longitudinal data that was generated over several patient visits and for this feeds sequential DXA and metadata into a LSTM to predict 10-year mortality probability. The study population consists of a cohort of ~3000 participants from the Health, Aging and Body Composition Study. For both approaches the study finds small but significant improvements in the prediction of 10-year mortality probability when adding DXA data to the known risk factors. Overall, the study confirms the well-known fact that body tissue composition is a predictor of mortality by using DXA to assess body tissue composition. However, the observed improvement is quite small and likely not clinically relevant. Moreover, the study lacks an investigation and understanding which information from the DXA scans was actually relevant as the whole image analysis part was more or less treated like a black-box. Here at least Grad-CAM visualizations or more advanced XAI techniques could have been applied to shed light on these relationships. Also technically the title of the study is wrong or at least misleading because the study in fact did not prove that it where body composition changes that improved mortality prediction.

The discussion is very brief, contains many unsubstantiated claims, does not address the limitations of the study, and does not place the study's findings in the context of the current state of the science on this topic. The discussion should be substantially revised.

Specific comments:

Abstract

The Abstract does not offer an overview over the study. Crucial information such as the author's motive, study design, results and conclusion must be included.

Introduction

The introduction is unfocused and distractive. Please consider to restructure (suggestion below):

a. What is the hypothesis from which you decided to conduct this study?

b. What is the relevance of your study?

1. Current methods to measure body composition and their limitations.

2. Ways intended to improve this method with the offered idea.

c. What is the originality of your study?

1. Similar research conducted in this field.

2. What is the advantage of your technique in contrast to established techniques?

Formulate a precise hypothesis and objective(s) of your study at the beginning of the introduction so that readers can follow your track of thinking.

P3 "To test these two hypotheses, we analyzed serial (...) be seen in Figure 2a." -> Belongs in the

Methods section

Methods

P8, paragraph "Study population ": Why did you decide to set this age range?

P9, paragraph "Image acquisition": Explain on how images were obtained. Positioning, image acquisition time, mode, dose, etc. .

P9, paragraph "Data preparation and split": Such things as python can be cited: citebay.com/how-to-cite/python/

P9, paragraph "Data preparation and split", "The dataset was then split (...) the final model comparison": Why did you decide to apply this specific split?

P9, last paragraph "Using custom software developed by the authors in Python": Please provide details.

Results

Numerous parts throughout this section are written in a commentary manner which should only find their place in the discussion.

P4, "Figure 1a shows (...) model inputs.": First of, you should explain the reason/results over which figure 1a is about. You are starting without any introduction.

P4, "...metadata: a mix of...": be specific about the metadata.

P5, "We acknowledge that it (...) of single-record models.": Not part of your results. Move to discussion.

P5, "Figure 4b demonstrates (...)": Same problem as with figure 1a, see above. Please clarify.

P5, "Interestingly, the models that (...) the longer sequences.": How do they profit?

P5, "...analyzing subgroup performance can also shed light ... models for these groups outright.": Not part of your results. Move to discussion.

P6, "... this is in line with our initial hypothesis that ... to be more pronounced in men than in women.": Not part of your results. Move to discussion.

P6, "... but while these methods are easy to use, they do not necessarily find the most relevant features.": Not part of your results. Move to discussion

P6, " this is just the performance of the metadata model as shown in Figure 1c.": Unclear.

P6, " Figure 2a depicts an attribution mask (...) factors for mortality.": Please revise the whole paragraph. It is hard to understand. Some parts have to be placed in the discussion section.

Discussion

P7, " This study demonstrates that TBDXA can supplement known mortality risk factors, and that changes in body composition over time can be a stronger predictor of mortality than any observations collected during a single visit.": How in particular can it supplement these factors? There is no clear link to the presented results.

P7, "the deep learning can effectively extract this information to improve prediction of all-cause mortality.": "the deep learning"? How does it improve mortality prediction?

P7, " The sequence model performance confirms our hypothesis that change in body composition over time more strongly predicts mortality than a single patient record.": Language. That seems rather obvious.

P7, " This suggests that beyond what is reflected in tabular risk factors, diffuse (or disseminated) changes in tissue, rather than one area such as central abdominal adiposity, contribute to biological processes that predispose to death.": This claim/thought has to be set in context of current literature. Also, please elaborate on how that relationship might be explained.

P8, " In both scenarios the models also performed especially well on the subgroup of the study

population that died from cardiovascular causes.": Language. Please elaborate on that claim.

P8, " As hypothesized, changes in body composition visible in TBDXA scans are associated with biological processes, such as inflammation, cellular senescence, and insulin resistance, that are particularly strong predictors of cardiovascular disease and cardiovascular mortality.": This

claim/thought has to be set in context of current literature. Also, please elaborate.

P8, " which reinforces the hypothesis that it is specifically change over time as represented by the TBDXA scans which contains relevant information for mortality prediction.": This sentence makes no sense or at least lacks substantial, further explanation.

P8: What are the studies limitations?

Reviewer #2 (Remarks to the Author):

Glaser and colleagues present a design of a deep neural network architecture, trained and evaluated on longitudinal DXA data from the Health, Aging and Body Composition (Health ABC) study to build a mortality risk prediction model. The authors hypothesized that observed changes in body composition by DXA would improve mortality prediction. The overall sequential model achieved an under the receiver operator characteristic curve of 0.79. The approaches described in this paper are important however the idea that more time points are better than a single time point is not novel and has several examples in the literature. The authors may wish to compare what has been accomplished here with the dataset size to what has been done for other imaging datasets. The strengths of the paper lie in the transparency and detail describing the AI architecture. General and specific comments are provided below.

General Comments

The purpose of the study is important, and the reviewers acknowledge paucity of data and literature on the topic of neural networks in the context of body composition and prediction of longevity. The manuscript requires revisions for clarity; general comments are enumerated in these paragraphs. The paper's current layout is confusing for this reviewer. Recommend rearranging the paper so that the workflow steps presented in the results match the flow presented in the discussion and methods sections.

This paper has many strengths including the sample size, repeat scans, and clear explanation of feature extraction from the scans. Some limitations that should be addressed include the lack of generalizability and insufficient information provided on the training sample related to the DXA scan and whether all participants fit entirely within the DXA field of view (FOV). This has important implications for accuracy of the training dataset and subsequent developed algorithm. A training dataset cannot include persons whose body sizes extended beyond the DXA FOV.

Specific Comments

Abstract

1. Page 3, please state aims /hypotheses that are presented in the introduction of the paper and subsequently report the results to better represent the conclusion of the abstract.

Introduction: Overall the introduction requires revision to better focus the manuscript. Additional information on the role of DXA images in prediction of all-cause mortality is warranted. Many of the papers cited as background are CT scans.

1. Page 3: the authors put forth two hypotheses: a) total body DXA will add predictive power to all-cause mortality models and b) observed changes in body composition will improve mortality prediction. In the last line of second paragraph, the authors state they hypothesize that models

using scans collected over time will outperform models from a single visit. As written, this is not a novel concept, and this sentence seems to serve more as rationale.

2. Page 3: The authors should include a sentence on advantage of employing LSTM over traditional AI for images or an architecture like boosted ensemble algorithm. This will further bolster rationale for hypothesis 2 while providing background for readers familiar with DXA, but unfamiliar with AI. Results: In some areas of the results section, there is text that would be more appropriately placed in the discussion. The Results section should only report model performance. Furthermore, subheadings should not be conclusive statements. As currently written, the results and discussion sections are hard to follow and not in sync for the information the authors are attempting to present.

1. Does the sentence on page 4: "These models are evaluated on all available datapoints for each participant represented as chronologically ordered sequence, where the predicted 10-year mortality is based on the most recent datapoint collected..." match the text included in data preparation and split? Rearranging the manuscript will benefit readers' understanding of the presented work.

2. Page 4, last two sentences: is this a fair comparison and is it truly underestimation or simply a temporal function? ie, how much time is allowable to denote fairness? Or does this simply support the use of longitudinal data over cross sectional for body composition variables?

3. Page 5, in the results, please report the AROUCs, and discuss the magnitude of change in the Discussion.

4. Page 6, the last paragraph of the results should be placed in the discussion

5. Page 6, the ablation study is a strength of this paper and context of results should be moved to discussion

Discussion: The discussion section requires revision to further expand upon the limitations of AI application and assumptions made in these approaches. Additionally, context of results and conclusive statements contained in the results section should be moved to the discussion section. An example of this is on page 7 where the authors discuss implications of higher VAT/central adiposity on mortality risk.

Methods: The reviewer strongly suggests rearranging sections to match how information is presented in results.

1. Study Population: Page 8, please use ethnicity/race terms consistently throughout manuscript. In some places the authors use African American, and others black. Also, information regarding ethnicity (Hispanic/non-Hispanic) should be provided

2. Did the authors perform any clustering analyses with features to investigate further their predictive power? For example, outside of CVD/cancer, were there other chronic disease diagnoses (ie, FBG, insulin cluster for T2DM) that could explain some of the variability in the models? The authors mention on page 8,-discussion, other biological processes and the reviewer suggests expanding upon this in addition to the ablation study.

3. Image Acquisition: Page 9, please provide information on DXA scan protocol regarding field of view

4. Data preparation and split: Page 9, Please provide another sentence on detail for determining which data were included in the dataset or rationale for how missing data were treated.

5. Has the custom software developed by the authors been previously published in peer-reviewed literature?

6. Single-record model architectures and training: Page 10, the authors do a thorough job of explaining the architecture, optimizers and addressing overfitting.

7. Sequence model architectures and training: Page 11, the authors describe this well, however a sentence related to data integration would be beneficial for readers.

Figures and Tables:

Please move Table S1 to main text as Table 1.

Please provide computational workflow Figure 3 as Figure 1

Figure 2a: Images are unclear, please provide more instruction for reader (similar to what is shown in Fig 3 and what is presented in text with regard to VAT)

We thank the reviewers for their thorough and thoughtful points. We have addressed each concern and provide an enumerated point by point response below.

Reviewer #1

The present study "DEEP LEARNING IDENTIFIES BODY COMPOSITION CHANGES OVER TIME IN TOTAL-BODY DXA IMAGING TO PREDICT

ALL-CAUSE MORTALITY" aims to demonstrate that the assessment of body composition by DXA can improve mortality risk prediction of a deep neural network. For this the authors use two approaches. The first approach, the single record model, aims to predict 10-year mortality probability from a baseline DXA scan that is processed with a convolutional neural network (DenseNet121) and after global average pooling gets concatenated with the metadata-model, a small network that processes known risk factors of 10-year mortality. The second approach takes advantage of longitudinal data that was generated over several patient visits and for this feeds sequential DXA and metadata into a LSTM to predict 10-year mortality probability. The study population consists of a cohort of ~3000 participants from the Health, Aging and Body Composition Study. For both approaches the study finds small but significant improvements in the prediction of 10-year mortality probability when adding DXA data to the known risk factors.

1. Overall, the study confirms the well-known fact that body tissue composition is a predictor of mortality by using DXA to assess body tissue composition. However, the observed improvement is quite small and likely not clinically relevant.

We thank you for those comments. We feel there are some more nuances to our results that should be mentioned here. The novelty of this study is in using imaging data directly to predict mortality, rather than deriving any particular measure of interest from the image. Previous literature using measures extracted from DXA, e.g. lean and fat mass derived from DXA, have shown no statistically significant association with mortality. Another novelty is using a sequence of longitudinal data that includes imaging directly. The improvement from the metadata model (0.69), whose inputs are most similar to what current mortality models would use, to our best performing model (0.79) is 0.10 AUROC.

2. Moreover, the study lacks an investigation and understanding which information from the DXA scans was actually relevant as the whole image analysis part was more or less treated like a black-box. Here at least Grad-CAM visualizations or more advanced XAI techniques could have been applied to shed light on these relationships.

We thank the reviewer for the suggestion. We added Grad-CAM visualization and expanded on the subsection elaborating on the integrated gradients and Grad-CAM feature maps.

3. Also technically the title of the study is wrong or at least misleading because the study in fact did not prove that it where body composition changes that improved mortality prediction.

We have updated to title To Be, "Deep Learning Predicts All-Cause Mortality from Longitudinal Total-Body DXA Imaging".

4. The discussion is very brief, contains many unsubstantiated claims, does not address the limitations of the study, and does not place the study's findings in the context of the current state of the science on this topic. The discussion should be substantially revised.

We thank the reviewer for the comment. The discussion has been extensively revised in line with the suggestions.

Specific comments:

Abstract

5. The Abstract does not offer an overview over the study. Crucial information such as the author's motive, study design, results and conclusion must be included.

Although we didn't use a structure abstract, it now has the key elements mentioned that are easily identifiable in the abstract.

6. The introduction is unfocused and distractive. Please consider to restructure (suggestion below):

We thank the reviewer for the extensive suggestions. The introduction was revised to be clearer

- a. What is the hypothesis from which you decided to conduct this study?

We revised the introduction to clearly state and highlight the hypotheses. We hypothesize that "1) TBDXA imaging data can add predictive power to all-cause mortality models and that 2) models using TBDXA scans collected over multiple visits for an individual will outperform models that use information from only a single visit."

- b. What is the relevance of your study?

- i. Current methods to measure body composition and their limitations.

We added an overview over the current state of the literature on the topic along with the shortcomings that our method seeks to address. We identify two potential areas of improvement: current models not being unable to directly use imaging data and thereby discarding potentially valuable information contained in the image, and current methods largely ignoring longitudinal data in favor of baseline data.

- ii. Ways intended to improve this method with the offered idea.

Our method is directly aimed at the identified shortcomings. Including TBDXA directly overcomes the limitation of having to extract information from imaging rather than using it directly, and using an LSTM to look at a sequence of datapoints directly leverages available longitudinal information.

- c. What is the originality of your study?

We are the first to show that including full TBDXA scans can improve mortality prediction. We are also the first to use LSTMs to directly integrate longitudinal imaging with traditional risk factors to improve mortality prediction.

- i. Similar research conducted in this field.

We added a more information exploring the current literature on mortality prediction from body composition as well as using similar deep learning approaches in medical imaging. We also better contextualize our results within the literature in the discussion

- ii. What is the advantage of your technique in contrast to established techniques?

We are able to include longitudinal data to improve predictions. Our method is also able to take advantage of a wide array of diverse information, notably TBDXA imaging with minimal preprocessing and no extra feature extraction necessary. Our method also doesn't require splitting the dataset on gender, as is commonly done in other methods (e.g. Newman 2003).

7. Formulate a precise hypothesis and objective(s) of your study at the beginning of the introduction so that readers can follow your track of thinking.

We thank you for your comments and have restructured the introduction as you suggest stating the following explicit hypotheses:

"We formulate two hypotheses:

Hypothesis 1: *Risk features derived from raw total-body DXA imaging data using deep learning are predictive of all-cause mortality with and without clinical risk factors.*

[...] We also hypothesize that:

Hypothesis 2: *Sequential TBDXA scans and recurrent neural network models (RNNs) outperform comparable models using only one observation with and without clinical risk factors. "*

8. P3 "To test these two hypotheses, we analyzed serial (...) be seen in Figure 2a." -> Belongs in the Methods section

The introduction was restructured to remove this part.

Methods

9. P8, paragraph "Study population ": Why did you decide to set this age range?

This is a retrospective analysis of a large and well described cohort. The age range was set by the HABC study with the intent of following a cohort of older adults who were free from mobility limitations at baseline during the critical period from 70-79 (<https://healthabc.nia.nih.gov/>). In future work, we plan on applying this model to a younger population.

10. P9, paragraph "Image acquisition": Explain on how images were obtained. Positioning, image acquisition time, mode, dose, etc. .

We added a brief description of the DXA acquisition protocol with a link to the detailed protocol: https://healthabc.nia.nih.gov/sites/default/files/dxa.om1_0.pdf

11. P9, paragraph "Data preparation and split": Such things as python can be cited: citebay.com/how-to-cite/python/
Thank you for the suggestion. We have updated our python reference as suggested by the citebay article.

12. P9, paragraph "Data preparation and split", "The dataset was then split (...) the final model comparison": Why did you decide to apply this specific split?

Given our relatively small data set size for a DL model, this split ensured that we got the maximum amount of training data while still maintaining a representative test set and a validation set large enough to reasonable choose hyperparameters. We embellished the section to explain this.

13. P9, last paragraph "Using custom software developed by the authors in Python": Please provide details.

We now provide more detail on P15. Also, a link is given to our code in the Code Availability section.

Results

14. Numerous parts throughout this section are written in a commentary manner which should only find their place in the discussion.

This section has been revised substantially to remove any commentary-style writing and just focus on presenting the results.

15. P4, "Figure 1a shows (...) model inputs.": First of, you should explain the reason/results over which figure 1a is about. You are starting without any introduction.

This has been reworked across all instances where it occurred in the text.

16. P4, "...metadata: a mix of...": be specific about the metadata.

Thank you for pointing out this needed more specifics. A detailed breakdown is now provided on page 14 in the methods section.

17. P5, "We acknowledge that it (...) of single-record models.": Not part of your results. Move to discussion.

We moved as suggested.

18. P5, "Figure 4b demonstrates (...)": Same problem as with figure 1a, see above. Please clarify.

Thank you for the suggestion. This has been reworked.

19. P5, "Interestingly, the models that (...) the longer sequences.": How do they profit?

This part of the analysis has been revised substantially to be more theoretically sound and reflect new insights into the model performance.

20. P5, "...analyzing subgroup performance can also shed light ... models for these groups outright.": Not part of your results. Move to discussion.

We moved as suggested.

21. P6, "... this is in line with our initial hypothesis that ... to be more pronounced in men than in women.": Not part of your results. Move to discussion.

We moved as suggested.

22. P6, "... but while these methods are easy to use, they do not necessarily find the most relevant features.": Not part of your results. Move to discussion

We moved as suggested.

23. P6, " this is just the performance of the metadata model as shown in Figure 1c.": Unclear.

Thank you for pointing this out. We have updated the section to include the following on page 7:

"It is also worth noting that while omitting the image as an input to the model is not explicitly included in the ablation results, the resulting model would be identical to the metadata model whose performance is detailed in Figure 1c."

24. P6, " Figure 2a depicts an attribution mask (...) factors for mortality.": Please revise the whole paragraph. It is hard to understand. Some parts have to be placed in the discussion section.

This paragraph has been revised, with and extended analysis now being located in the discussion.

Discussion

25. P7, " This study demonstrates that TBDXA can supplement known mortality risk factors, and that changes in body composition over time can be a stronger predictor of mortality than any observations collected during a single visit.": How in particular can it supplement these factors? There is no clear link to the presented results.

Please see our response to point 26 below.

26. P7, "the deep learning can effectively extract this information to improve prediction of all-cause mortality.": "the deep learning"? How does it improve mortality prediction?

Our first hypothesis is that there is information present in the TBDXA imaging pertaining to mortality that is not captured by the metadata. for example localized body composition information, bone shape relating to degeneration, and distribution of lean and adipose tissue. The DL models see both the the metadata variables and the DXA scans, so the model can learn to extract information from the scans that complements the metadata variables. We show that this improves overall mortality prediction. We have updated P7 to clarify our logic.

27. P7, " The sequence model performance confirms our hypothesis that change in body composition over time more strongly predicts mortality than a single patient record.": Language. That seems rather obvious.

We have updated several sections of the paper to set this statement into its proper context within mortality literature. While it might be qualitatively obvious, it is not obvious how to quantify this relationship in a statistical model. Previous attempts in the literature have been unsuccessful, with the use of longitudinal information resulting in worse model performance compared to baseline information. This is presumably due to overfitting in the more complex models. More information on this is provided as part of the introduction and discussion.

28. P7, " This suggests that beyond what is reflected in tabular risk factors, diffuse (or disseminated) changes in tissue, rather than one area such as central abdominal adiposity, contribute to biological processes that predispose to death.": This claim/thought has to be set in context of current literature. Also, please elaborate on how that relationship might be explained.

This has been revised for clarity. There is no precedent in the literature because no modeling has been done using full medical imaging for body-composition-based all-cause mortality, much less in tandem with traditional risk factors. This is something that has to be explored in future analyses. Other examples in the literature (see page 3, in the introduction) have repeatedly shown that using medical imaging instead of traditional risk factors for mortality prediction can be beneficial, however, these examples are usually restricted to specific populations and not general body-composition-based approaches. Our results in combination with the feature maps do however show that the models robustly identify mortality signals across the entire scan without disproportionately focusing on a single area, supporting the statement that diffuse information is being used.

29. P8, " In both scenarios the models also performed especially well on the subgroup of the study population that died from cardiovascular causes.": Language. Please elaborate on that claim.

This discussion on this observation has been extended:

"[...] the models performed most strongly on the group with cardiovascular disease. In fact, every single model achieved the highest AUROC observed for any subgroup of the testing data for the participants who died of cardiovascular disease. This observation aligns well with the existence of body-composition-related factors contributing to cardiovascular disease, information that would be available to the model through both the TBDXA and metadata variables."

30. P8, " As hypothesized, changes in body composition visible in TBDXA scans are associated with biological processes, such as inflammation, cellular senescence, and insulin resistance, that are particularly strong predictors of cardiovascular disease and cardiovascular mortality.": This claim/thought has to be set in context of current literature. Also, please elaborate.

This paragraph has been revised substantially.

31. P8, " which reinforces the hypothesis that it is specifically change over time as represented by the TBDXA scans which contains relevant information for mortality prediction.": This sentence makes no sense or at least lacks substantial, further explanation.

This has been removed as it is hard to conclusively prove this claim.

32. P8: What are the studies limitations?

A paragraph for the study's limitations has been added.

Reviewer #2:

Glaser and colleagues present a design of a deep neural network architecture, trained and evaluated on longitudinal DXA data from the Health, Aging and Body Composition (Health ABC) study to build a mortality risk prediction model. The authors hypothesized that observed changes in body composition by DXA would improve mortality prediction. The overall sequential model achieved an under the receiver operator characteristic curve of 0.79.

33. The approaches described in this paper are important however the idea that more time points are better than a single time point is not novel and has several examples in the literature. The authors may wish to compare what has been accomplished here with the dataset size to what has been done for other imaging datasets.

We revised the manuscript to include more related work in the introduction, including work that derives input features from longitudinal information and work that uses imaging data for all-cause mortality prediction. However, our approach is unique in the field of body-composition-based mortality prediction in that we do not engineer input features to represent longitudinal information (for instance by subtracting the current-most value for a variable from its baseline) but rather use deep learning to directly model the time dependency. This is clarified on page 3 of the introduction.

The strengths of the paper lie in the transparency and detail describing the AI architecture. General and specific comments are provided below.

General Comments

34. The manuscript requires revisions for clarity; general comments are enumerated in these paragraphs.

The manuscript has been revised substantially, restructuring the introduction, results, and discussion sections to create a more linear and cohesive text.

35. The paper's current layout is confusing for this reviewer. Recommend rearranging the paper so that the workflow steps presented in the results match the flow presented in the discussion and methods sections.

We thank the reviewer for their comment, the text was revised with this arrangement in mind.

36. Some limitations that should be addressed include the lack of generalizability and insufficient information provided on the training sample related to the DXA scan and whether all participants fit entirely within the DXA field of view (FOV). This has important implications for accuracy of the training dataset and subsequent developed algorithm. A training dataset cannot include persons whose body sizes extended beyond the DXA FOV.

We thank the reviewer for bringing the issue of FOV to our attention. Fortunately, we believe that with our proposed method this does not present an issue. Because the model is trained on random slices and evaluated only on a center crop, it never actually “sees” the full TBDXA slice. In cases where participants don’t fully fit within the FOV, our model would only see the center crop at test time anyway, being unaware of any parts of the body that might extend beyond the DXA FOV.

Specific Comments

Abstract

37. Page 3, please state aims /hypotheses that are presented in the introduction of the paper and subsequently report the results to better represent the conclusion of the abstract.

The abstract has been completely revised to better address these concerns.

Introduction

38. Overall the introduction requires revision to better focus the manuscript. Additional information on the role of DXA images in prediction of all-cause mortality is warranted. Many of the papers cited as background are CT scans.

The introduction has been revised substantially. We have elaborated on the choice of TBDXA imaging specifically here:

“TBDXA imaging was selected specifically because the scans contain rich body composition information such as central adiposity, lean tissue mass, and bone density. Additionally, DXA scans are relatively cheap, can be taken at any age, and only take a low radiation dose making them well-suited to collect longitudinal data with. It is also easier to collect total-body scans with DXA than with other modalities that would otherwise be good candidates for this type of analysis.”

39. Page 3: the authors put forth two hypotheses: a) total body DXA will add predictive power to all-cause mortality models and b) observed changes in body composition will improve mortality prediction. In the last line of second paragraph, the authors state they hypothesize that models using scans collected over time will outperform models from a single visit. As written, this is not a novel concept, and this sentence seems to serve more as rationale.

This issue has been raised by the other reviewers as well. We have revised the text for clarity on the subject. Specifically, we have updated several sections of the paper to set this statement into its proper context within mortality literature. While it might seem intuitively obvious, a lot of previous literature has not been able to confirm this intuition and utilizing longitudinal information has often been shown to result in worse performing models when compared to simple baseline information. More information on this is provided as part of the introduction and discussion.

40. Page 3: The authors should include a sentence on advantage of employing LSTM over traditional AI for images or an architecture like boosted ensemble algorithm. This will further bolster rationale for hypothesis 2 while providing background for readers familiar with DXA, but unfamiliar with AI.

We thank the reviewer for the suggestion. We included the following information on why an LSTM is preferential for the task:

“Using a LSTM-based model has two major benefits: the model’s ability to process sequences of different lengths without requiring padding or truncating any of the data, which is desirable because not all participants have the same number of datapoints available for various reasons; and recurrent neural network architectures make the assumption of temporal dependence between the samples within a sequence, which is not inherently the case with many alternative methods like ensemble models. “

Results

41. In some areas of the results section, there is text that would be more appropriately placed in the discussion. The Results section should only report model performance. Furthermore, subheadings should not be conclusive statements. As currently written, the results and discussion sections are hard to follow and not in sync for the information the authors are attempting to present.

The text has been revised to better respect these suggestions. Headings have been made more homogenous across sections.

42. Does the sentence on page 4: "These models are evaluated on all available datapoints for each participant represented as chronologically ordered sequence, where the predicted 10-year mortality is based on the most recent datapoint collected..." match the text included in data preparation and split? Rearranging the manuscript will benefit readers' understanding of the presented work.

We revised this for clarity and to better align across sections.

43. Page 4, last two sentences: is this a fair comparison and is it truly underestimation or simply a temporal function? ie, how much time is allowable to denote fairness? Or does this simply support the use of longitudinal data over cross sectional for body composition variables?

This has been revised for more clarity on the subject. What this observation mostly supports is the use of longitudinal data over cross-sectional data in our specific modeling context, this is not necessarily the case for other approaches and detailed in the revised introduction of the paper. This added paragraph details our rationale:

"The intent of this adjustment to match the test sets for the sequence and single-record models as closely as possible. By using only the most recent scan for each participant for the single-record models, we ensure that 10-year mortality labels are the same for both models across the test set, whereas otherwise the same participant can have their label change throughout the course of the study if they died within follow-up but more than ten years after their baseline scan."

44. Page 5, in the results, please report the AROUCs, and discuss the magnitude of change in the Discussion.

The manuscript has been rearranged to address this.

45. Page 6, the last paragraph of the results should be placed in the discussion

We moved as suggested.

46. Page 6, the ablation study is a strength of this paper and context of results should be moved to discussion

We moved as suggested.

Discussion:

47. The discussion section requires revision to further expand upon the limitations of AI application and assumptions made in these approaches. Additionally, context of results and conclusive statements contained in the results section should be moved to the discussion section. An example of this is on page 7 where the authors discuss implications of higher VAT/central adiposity on mortality risk.

This has been adjusted and a limitations subsection has been added.

Methods:

48. The reviewer strongly suggests rearranging sections to match how information is presented in results.

Thank you for noting this. We have rearranged the results to match the methods section.

49. Study Population: Page 8, please use ethnicity/race terms consistently throughout manuscript. In some places the authors use African American, and others black. Also, information regarding ethnicity (Hispanic/non-Hispanic) should be provided.

We thank the reviewer for pointing this out to us, we fixed the use of race term and detailed ethnicity information.

50. Did the authors perform any clustering analyses with features to investigate further their predictive power? For example, outside of CVD/cancer, were there other chronic disease diagnoses (ie, FBG, insulin cluster for T2DM) that could explain some of the variability in the models? The authors mention on page 8,-discussion, other biological processes and the reviewer suggests expanding upon this in addition to the ablation study.

Thank you for your suggestion. We did perform AUC analysis by BMI groups and by diabetes status. We also did further sequence AUC analysis associated with the Kaplan-Meier curves we put into the supplemental material for significant meta variables (i.e. (percent) weight change, adl status change, walking speed change, change in diabetes status, bmi group change).

51. Image Acquisition: Page 9, please provide information on DXA scan protocol regarding field of view

Please see response to point 9 above.

52. Data preparation and split: Page 9, Please provide another sentence on detail for determining which data were included in the dataset or rationale for how missing data were treated.

In response to your suggestion, we updated to the following: Original: "Since questionnaire and exam measures were irregularly scheduled and did not always line up with each other or when imaging was done, not all data was collected contemporaneously with the scans. ~~Which datapoints were included in our dataset was determined by when scans were collected. Missing values in other modalities were then simply backfilled with the most recent values available or a default out-of-distribution value.~~" Input records were constructed based on when scans were collected. For each available scan, metadata from the same year was used to construct a record. If some data was not collected that year, that information was filled in with the most recently collected value. If that data had not been collected before the scan was taken, an out-of-distribution filler value was used. For years where metadata was collected but no scan, no extra records were constructed.

53. Has the custom software developed by the authors been previously published in peer-reviewed literature?

The software was created for this analysis and this is the first time it has been presented.

54. Single-record model architectures and training: Page 10, the authors do a thorough job of explaining the architecture, optimizers and addressing overfitting.

55. Sequence model architectures and training: Page 11, the authors describe this well, however a sentence related to data integration would be beneficial for readers.

We added information on how the LSTM integrates data in the introduction as detailed in response to point 39.

Figures and Tables:

56. Please move Table S1 to main text as Table 1.

The table has been moved to the main text as table 6 to reflect the order in which it appears in the text.

57. Please provide computational workflow Figure 3 as Figure 1

The order has been chosen to reflect the order in which figures are referenced in the manuscript.

58. Figure 2a: Images are unclear, please provide more instruction for reader (similar to what is shown in Fig 3 and what is presented in text with regard to VAT)

The sections on saliency maps have been revised with updated information. The text now includes more information on what the maps are intended to show and how to interpret them along with some analysis of the visualizations in the discussion.

Reviewers' comments:

Reviewer #1 (Remarks to the Author):

After extensive revisions, some of the shortcomings have been improved. However, some revisions are necessary to better stress the results and their relevance in the context of current knowledge.

1. Still, the manuscript lacks structure, a clear sub-division of the different paragraphs would improve the flow of reading. The different chapters should therefore be divided with more specific subheadings.
2. Especially the abstract, but also every other chapter except the discussion should not be written in a "narrative style", the use of pronouns such as "we, our, us" should be avoided.
3. The study confirms that body composition measured via DXA is relevant for mortality prediction. However, to really test the hypotheses, the study collective needs to be itemized regarding their diagnoses and causes of death to further check for biases in the models' predictions. A table which lays out the study population regarding their clinical background would improve the scant specification which is given in the paragraph "study population". This is especially relevant, when a reference to "clinical risk factors", as you did in hypothesis 2, is made.
4. Formulations like "elsewhere" (l.319) or a reference to the systems manual (l. 331) should be specified and avoided.
5. Please discuss what position DXA-scans have in clinical routine care. Wouldn't a network trained on CT-imaging bring more improvement for clinicians as CT-scans are performed way more than DXA-scans and for multiple indications, especially in elderly patients? Referring to MRI-scans (l. 74) does not sufficiently support your claim.
6. The discussion needs further revision. Claims such as "(...) our deep learning approach does not require this additional processing step while (...)" must be explained.
7. In lines 220f. you conclude that "Overall, this study demonstrates that risk factors derived from TBDXA using deep neural networks can supplement known mortality risk factors, and that changes in body composition over time can be a stronger predictor of mortality than any observations collected during a single visit.". However, this claim needs to be discussed more extensively: in which regard is this expectable finding relevant for your approach or how does your approach offer improvement to the described problem?
8. Please structure the section "feature importance" (l. 241f.) more clearly.
9. "The saliency maps in Figure 4 are intended to shed light on" (l. 257): narrative style.
10. In lines 257 following you begin to analyze what the DNN activation maps might bring to light. It would be of interest to further analyze what parts might be really "biologically meaningful" here. Deriving precise conclusions from these visualizations might be "difficult" but even more relevant as these could suggest probable directions for future research. The activation maps in F4 seem to highlight specific muscles, visceral tissues, and also adipose tissues in quite clearly definable regions. Please elaborate on this interesting point.
11. In line 296 you state that "(...) this subtlety must be considered when evaluating any model of 297 longitudinal data.". To what extent is it considered in your study?

Reviewer #2 (Remarks to the Author):

None

We thank the reviewer for providing us with more thorough and thoughtful feedback. We have addressed each concern and provide a point-by-point response below.

Reviewer #1:

After extensive revisions, some of the shortcomings have been improved. However, some revisions are necessary to better stress the results and their relevance in the context of current knowledge.

1. Still, the manuscript lacks structure, a clear sub-division of the different paragraphs would improve the flow of reading. The different chapters should therefore be divided with more specific subheadings.

We have divided the results and discussion sections with more specific subheadings.

2. Especially the abstract, but also every other chapter except the discussion should not be written in a “narrative style”, the use of pronouns such as “we, our, us” should be avoided.

This has been corrected throughout the manuscript. All uses of personal pronouns outside of the discussion have been removed.

3. The study confirms that body composition measured via DXA is relevant for mortality prediction. However, to really test the hypothesized, the study collective needs to be itemized regarding their diagnoses and causes of death to further check for biases in the models’ predictions. A table which lays out the study population regarding their clinical background would improve the scant specification which is given in the paragraph “study population”. This is especially relevant, when a reference to “clinical risk factors”, as you did in hypothesis 2, is made.

More detail on the clinical background has been added. In the study population paragraph, we added the following paragraph (l. 335 ff.):

At intake, participants’ medical characteristics were recorded through a questionnaire. Concerning this study, across the entire population, 58% self-reported a previous heart condition (one or multiple of: previous heart attack or myocardial infarction, a history of chest pain, previous congestive heart failure, previous strokes, a history of hypertension), 21% self-reported previous respiratory illness (one or multiple of: diagnosed with asthma, diagnosed with chronic bronchitis, diagnosed with emphysema, diagnosed with chronic obstructive respiratory disease (COPD), diagnosed with pneumonia in the past twelve months), 31% self-reported stomach or gallbladder issues (one or multiple of: previous stomach or duodenal ulcers, previous stomach or intestinal bleeding, previous surgery to remove parts of the stomach, gallstones), 15% self-reported being previously diagnosed with diabetes by a doctor, and 19% reported being previously diagnosed with cancer by a doctor.

Additionally, a table (Table S2) was added to the supplement detailing model performance for the four most common primary and underlying causes of death in the dataset respectively in line with the reviewer’s suggestion.

4. Formulations like “elsewhere” (l.319) or a reference to the systems manual (l. 331) should be specified and avoided.

These sentences of the Study Population and Image Acquisition subsections have been revised and replaced with more details of the participant recruitment and image acquisition procedures (l. 354 ff.). We kept the explicit reference to the study’s DXA operation manual because we anticipate that some readers will find this helpful:

[...] strict acquisition procedures in place to ensure reproducibility, including a detailed DXA operations manual, annual operator training, and contracting the services of a DXA reading center. Throughout the study, whole-body phantoms and human volunteers were used to verify proper DXA calibration.

5. Please discuss what positon DXA-scans have in clinical routine care. Wouldn’t a network trained on CT-imaging bring more improvement for clinicians as CT-scans are performed way more than DXA-scans and for multiple indications, especially in elderly patients? Referring to MRI-scans (l. 74) does not sufficiently support your claim.

Total body DXA is the criterion method for risk assessments that utilize total body composition such as bone density, and muscle and fat masses. It is broadly available throughout the world and the US for this purpose, is low dose and inexpensive compared to CT. CT is extensively used for diagnostic imaging, is higher dose, and is often not available for risk screening due to its use as an emergency imaging modality. Thus, our DXA modeling may be more useful to clinical practice than CT. However, it is a good idea to further explore CT for mortality using AI, especially for opportunistic screening, and we plan to explore this in the future.

We have updated the introduction and discussion to clarify these points (lines 71 ff. and 320 ff.).

We also included the recent review by Pickhardt on CT opportunistic screening and the following references on dose.

References:

Pickhardt, P. J. (2022). Value- Added opportunistic CT screening: state of the art. Radiology, 211561.

Shepherd JA, Ng BK, Sommer MJ, Heymsfield SB. Body composition by DXA. Bone. 2017; 104:101-5.

Blake, G., Naeem, M. & Boutros, M. Comparison of effective dose to children and adults from dual X-ray absorptiometry examinations. Bone. 38935–942 (2006).

6. The discussion needs further revision. Claims such as “(...) our deep learning approach does not require this additional processing step while (...)” must be explained.

This has been clarified in the text (l. 252 f.):

[...] our deep learning approach uses a single model for both groups. The model is flexible enough to learn differences between groups but can benefit from the larger training dataset that results from combining groups.

Other parts of the discussion, most notably the feature importance section, have also been further reworded as addressed in 8.

7. In lines 220f. you conclude that “Overall, this study demonstrates that risk factors derived from TBDXA using deep neural networks can supplement known mortality risk factors, and that changes in body composition over time can be a stronger predictor of mortality than any observations collected during a single visit.”. However, this claim needs to be discussed more extensively in which regard is this expectable finding relevant for your approach or how does your approach offer improvement to the described problem?

Two findings are discussed. The first is that using deep learning, we can create a more powerful mortality predictor by integrating raw TBDXA scans with traditional risk factors. We expanded the discussion addressing this (l. 210 ff.):

[...] our method of training a deep learning model to extract information from the raw TBDXA scan employs more powerful models that can extract information contained in TBDXA imaging to automatically derive complementary features to what is already represented in traditional risk factors. This end-to-end approach removes the reliance on crude summary statistics such as fat or lean mass which must be derived from images by a radiologist or segmentation algorithm.

The second finding is that our model can improve mortality prediction using longitudinal data. While this is expected in the sense that “more information is always better”, it is not obvious that a statistical model can learn this relationship from limited training data. We discuss why this is a non-trivial finding here (l. 220 ff.):

This observation is novel in the all-cause mortality literature as no such complicated multivariate model has been assessed on longitudinal data before our study. [...] In light of the results presented in this paper, this suggests that simple summary values derived from TBDXA such as total lean and fat mass may not be sufficient to represent the complicated longitudinal information contained in a sequence of DXA scans. [...] Our results also show a greater relative improvement from TBDXA inclusion in the longitudinal setting versus the single-record setting, suggesting that unlike the risk factors analyzed in Westbury et al.⁴, the raw imaging data may have more predictive power in a longitudinal context.

Effectively extracting task-relevant information from rich modalities such as imaging and even longitudinal imaging without overfitting to training data is a non-trivial task. It has not been demonstrated with traditional statistical models.

8. Please structure the section “feature importance” (l. 241f.) more clearly.

As suggested in 1., we added subheadings to provide more structure to the paragraph. We also changed the wording in both the ablation-study and saliency-map paragraphs for clarity and overall structure.

9. “The saliency maps in Figure 4 are intended to shed light on” (l. 257): narrative style.

This was reworded to (l. 275 ff.):

The saliency maps in Figure 4 highlight regions of the DXA scans that are of particular value for the mortality models’ predictions and can serve as a starting point when analyzing which parts of the TBDXA imaging drive the increase in mortality prediction performance.

10. In lines 257 following you begin to analyze what the DNN activation maps might bring to light. It would be of interest to further analyze what parts might be really “biologically meaningful” here. Deriving precise conclusions from these visualizations might be “difficult” but even more relevant as these could suggest probable directions for future research. The activation maps in F4 seem to highlight specific muscles, visceral tissues, and also adipose tissues in quite clearly definable regions. Please elaborate on this interesting point.

We agree that using end-to-end models to identify biologically meaningful risk factors is a tantalizing possibility worth exploring. But while “explainable AI” methods such as GradCAM can suggest features of interest for further study, these methods do not enable any concrete conclusions to be drawn. The model is making use of multiple sources of information and isolating the contributions of each of these sources will require a larger dataset. We plan to address this issue in follow-up work.

While addressing point 8., parts of this section have been reworded for clarity.

Specifically, we reworded to (l. 277 f.):

While it is ~~difficult~~ not sensible to derive precise conclusions from these visualizations, they do grant some insight into deep convolutional models.

Within this paragraph, some parts suggest potential direction for future research, such as (l. 292 f.):

Many of these maps [...] show heightened activation around the periphery of the body, perhaps identifying body shape or subcutaneous fat distribution.

And (l. 395 ff.):

[...] these maps seem to suggest that a lot of the information derived from scans to complement the metadata might be localized body composition information, distribution of lean and adipose tissue, or even bone shape and degeneration.

11. In line 296 you state that “(...) this subtlety must be considered when evaluating any model of 297 longitudinal data.”. To what extent is it considered in your study?

The problem of incomplete/noisy labels is addressed by performing the experiment in Figure 3b, which assesses the model performance on subgroups of subjects with different numbers of scans. Subjects with many scans are more likely to have incomplete follow-up, information that could potentially be exploited by any statistical model. Our results show that the sequence model shows small improvements over the cross-sectional model for most subgroups, and not just those subgroups with the most scans. From the text (l. 316 f.):

Our analysis in Figure 3b shows that the sequence model is more accurate than the single-record model even for the early years of the study where this effect is not a factor [...]

We highlight this in the discussion since it is an important consideration for the analysis of this study and design of future studies.

REVIEWERS' COMMENTS:

Reviewer #1 (Remarks to the Author):

In the latest revision, the authors have now addressed all my concerns. Thank you.